# Infection with a newly designed dual fluorescent reporter HIV-1 effectively identifies latently infected CD4[+] T cells

Jinfeng Cai[1,2], Hongbo Gao[1,2], Jiacong Zhao[1,2], Shujing Hu[1,2], Xinyu Liang[1,2], Yanyan Yang[1], Zhuanglin Dai[1], Zhongsi Hong[3], Kai Deng[1,2]*

[1]Institute of Human Virology, Key Laboratory of Tropical Disease Control of Ministry of Education, Zhongshan School of Medicine, Sun Yat-sen University, Guangzhou, China; [2]Department of Immunology, Zhongshan School of Medicine, Sun Yat-sen University, Guangzhou, China; [3]Department of Infectious Diseases, Fifth Affiliated Hospital, Sun Yat-sen University, Zhuhai, China

**Abstract** The major barrier to curing HIV-1 infection is a small pool of latently infected cells that harbor replication-competent viruses, which are widely considered the origin of viral rebound when antiretroviral therapy (ART) is interrupted. The difficulty in distinguishing latently infected cells from the vast majority of uninfected cells has represented a significant bottleneck precluding comprehensive understandings of HIV-1 latency. Here we reported and validated a newly designed dual fluorescent reporter virus, DFV-B, infection with which primary CD4[+] T cells can directly label latently infected cells and generate a latency model that was highly physiological relevant. Applying DFV-B infection in Jurkat T cells, we generated a stable cell line model of HIV-1 latency with diverse viral integration sites. High-throughput compound screening with this model identified ACY-1215 as a potent latency reversing agent, which could be verified in other cell models and in primary CD4[+] T cells from ART-suppressed individuals ex vivo. In summary, we have generated a meaningful and feasible model to directly study latently infected cells, which could open up new avenues to explore the critical events of HIV-1 latency and become a valuable tool for the research of AIDS functional cure.

*For correspondence:
dengkai6@mail.sysu.edu.cn

Competing interests: The authors declare that no competing interests exist.

## Introduction

Despite antiretroviral therapy (ART) efficiently blocks productive HIV-1 replication, viruses persist in a small pool of latently infected CD4[+] T cells, which are widely considered the major barrier to curing HIV-1 infection. These latently infected cells harbor transcriptional silent but replication-competent proviral genome, which was demonstrated to be the origin of rapid viral rebound after ART is interrupted (*Finzi et al., 1997*; *Chun et al., 2008*). Latently infected CD4[+] T cells are essentially indistinguishable from uninfected cells and are therefore not recognized by immune responses, including broadly neutralizing antibodies or cytotoxic T lymphocytes (CTLs). Current technologies also do not allow direct identification or purification of live latently infected cells from the infected individuals on ART. It is therefore vitally important to create novel and physiological relevant cell models of HIV-1 latency, which can be further exploited to study the mechanisms underlying HIV-1 latency and to develop novel strategies for AIDS functional cure.

The difficulty in distinguishing latently infected cells from the vast majority of uninfected cells has represented a major bottleneck precluding comprehensive understandings of HIV-1 latency. To tackle this question, researchers have tried to establish reporter virus systems to differentiate productively and latently infected cells. The reporter viruses usually incorporate two fluorescent proteins, with one dependent on LTR activity while the other under the control of a constitutive

promoter (*Calvanese et al., 2013*; *Dahabieh et al., 2013*; *Battivelli et al., 2018*; *Kim et al., 2019*). However, some apparent defects existed with the previously reported systems, especially when infecting primary CD4$^+$ T cells: (a) unclear cell subpopulations, for example, the boundaries of productively infected cells were obscured (*Calvanese et al., 2013*; *Battivelli et al., 2018*; *Kim et al., 2019*); (b) potential interference of cytokines on HIV-1 latency, as all infected cells (*Dahabieh et al., 2013*; *Battivelli et al., 2018*) were maintained in IL-2 containing medium for latency establishment; and (c) low frequency of reactivation of the identified latently infected cells, for instance, only 5% of HIV$_{GKO}$ cells were reactivated by bryostatin-1 and panobinostat (*Battivelli et al., 2018*), which bona fide limit its application to explore the mechanism of HIV latency maintenance and reactivation in vitro.

To improve on the cell model of HIV-1 latency, here we set up a newly designed dual fluorescent reporter virus, DFV-B, infection of which can distinguish latently infected from productively infected and uninfected cells. With DFV-B infection, we can sort out latently infected primary CD4$^+$ T cells, in which more than 50% can be reactivated by subsequent PMA and ionomycin stimulation. We also showed that DFV-B infection in Jurkat cells can generate a bunch of stable latently infected cells (named J-mC cells), which were obtained by natural infection, therefore circumventing the clonal-derived biases of some previous cell lines like J-Lat. We went on to show that the J-mC system is a robust model of HIV-1 latency, as a high-throughput compound screening with this model successfully identifies a novel latency reversal agent (LRA) ACY-1215, whose activity could be verified in other cell models and in primary CD4$^+$ T cells from ART-suppressed individuals. As an oral HDAC6-selective inhibitor, ACY-1215 has a higher safety and tolerability profile than the currently available pan-HDAC inhibitors such as SAHA or panobinastat, making it a promising LRA in future applications for HIV-1 functional cure.

## Results

### DFV-B infection directly identifies latently infected primary CD4$^+$ T cells

To specifically identify latently infected cells, we constructed a new reporter virus by engineering GFP and mCherry into the HIV-1 genome, in which GFP expression was controlled by LTR, while mCherry expression was independently controlled by the constitutive promoter EF-1α. This viral construct was named DFV-B (dual fluorescent virus-B) (*Figure 1A*). To test the infectivity of DFV-B, activated primary CD4$^+$ T cells from HIV-1 naïve individuals were infected with DFV-B incorporating a CXCR4-tropic env (*Figure 1B*). Productively infected (double-positive, DP) and latently infected (single mCherry-positive, mC) CD4$^+$ T cells were clearly distinguishable (*Figure 1B*), and the percentage of the DP cells was roughly 16 times more than the single mC-positive cells on day 3 post-infection (p.i.) in STCM (Super T cell medium) (*Figure 1B*). To mimic the process of infected activated CD4$^+$ T cells transitioning back to a more resting state, and monitor how the dynamics of productive and latent infection change during this process, DFV-B-infected cells were transferred to cytokine-free medium (CFM) on day 3 p.i. As culture time increased, the proportion of DP cells decreased significantly, while the proportion of mC cells gradually increased (*Figure 1C*). mC cells accounted for up to 30% of the total infected cells on day 10 p.i. (*Figure 1C*), suggesting some of the productively infected cells survived, reverted to a more resting state, and turned off viral gene (GFP) expression, which may add to the increased proportion of latently infected cells.

To validate the single mC-positive cells that were indeed latently infected, a series of experiments were performed as illustrated in *Figure 2A*. We sorted the DP, the single mC-positive, and the double-negative (DN) cells by flow cytometry at 7 days post DFV-B infection and validated the purity of each population (*Figure 2—figure supplement 1A*). The sorted mC cells were then cultured in CFM and activated by PMA and ionomycin for 3 days. During this process, the viability of mC cell population was more than 40% (*Figure 2—figure supplement 1B*). Compared to the DMSO control, over 50% of total cells were converted from mC to DP cells (*Figure 2B*), suggesting that more than half of the latently infected cells were reactivated. At the same time, DP and DN cells were not affected (*Figure 2B*). In addition, these sorted populations were analyzed for HIV-1 RNA and protein expression. As expected, the DP cells transcribed significantly higher amounts of HIV-1 mRNA, nearly 10 times as the mC cells (*Figure 2C*). HIV-1 Gag and Vif protein production were detected by western blot in DP cells, but not in mC cells (*Figure 2D*). To further verify that the absence of GFP

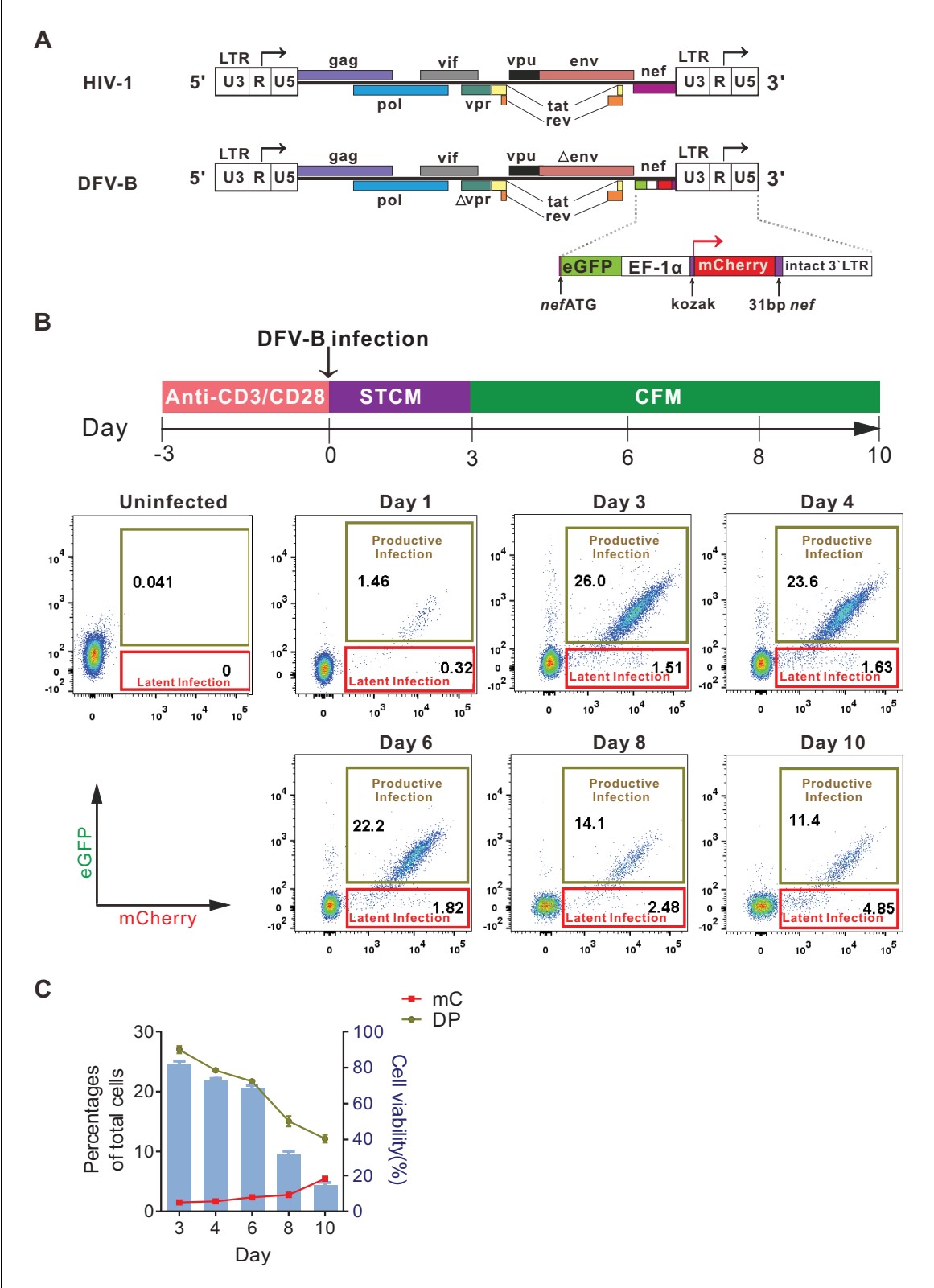

**Figure 1.** Infection with a newly designed dual fluorescent reporter HIV-1 identifies latently infected primary CD4+ T cells. (**A**) Diagram of the derivation of DFV-B from the original NL4-3 strain of HIV-1. Top panel depicts the genomic structure of HIV-1 NL4-3; bottom panel shows the genomic structure of DFV-B and the major modifications comparing to NL4-3. (**B**) Representative experiment of DFV-B infection in activated primary CD4+ T cells. Top panel depicts the scheme of experimental procedure. Isolated primary CD4+ T cells from HIV naïve individuals were stimulated with αCD3/αCD28

*Figure 1 continued on next page*

*Figure 1 continued*

antibodies in the presence of IL-2 and TCGF for 3 days, and the activated CD4$^+$ T cells were then infected with DFV-B incorporating a CXCR4-tropic Env at the MOI of 0.1 to 1 for 3 days in STCM and were subsequently transferred to CFM. Bottom panel shows the infection profiles of stimulated primary CD4$^+$ T cells by DFV-B at indicated time points. (C) Data from panel B are displayed graphically. Line charts show the percentages of latent infection (mC) and productive infection (DP). Histogram represents the cell viability. Data indicate mean ± SD from three different donors. TCGF: T cell growth factor; CFM: cytokine-free medium; STCM: super T cell medium, CFM with IL-2 and TCGF.

production was due to viral latency but not possible genomic deletion, we quantified the cell-associated DNA and RNA of fluorescent protein from different populations respectively. Comparing to the uninfected DN cells, both GFP-DNA and mCherry-DNA were comparable between mC and DP cells, indicating no difference in viral integration (*Figure 2E*). Meanwhile, levels of GFP mRNA in DP cells were much higher than that in mC cells, but mCherry mRNA levels remained the same (*Figure 2E*), consistent with the phenotypes observed by FACS. A small fraction of single GFP$^+$ cells existed after DFV-B infection (*Figure 1B*), and we traced and found that the proportion of single GFP$^+$ cells decreased from 0.7% to 0.2% (similar to uninfected cells) as the culture time increased (*Figure 2— figure supplement 1C*), which is significantly lower than the previous reported dual reporter viral systems (*Calvanese et al., 2013*; *Kim et al., 2019*). The integrated DNA levels of GFP and mCherry in this single GFP$^+$ cells were not different (*Figure 2—figure supplement 1D*), suggesting both reporters were still stably integrated. We speculated that this unlikely phenotype of single GFP$^+$ cells could be due to the differential dynamics of fluorescent protein degradation. Taken together, these results demonstrated that infection of DFV-B reporter virus in primary CD4$^+$ T cells could directly identify live latently infected cells.

## DFV-B infection confirms effector-to-memory transitioning (EMT) CD4$^+$ T cells are the primary targets for latency establishment

The latent HIV-1 reservoir mainly resides in resting memory CD4$^+$ T cells (*Chun et al., 1997*; *Finzi et al., 1997*; *Chomont et al., 2009*), but they are normally not the direct targets for infection, due to the blocks in reverse transcription (*Korin and Zack, 1998*), integration (*Bukrinsky et al., 1992*) and host restriction by SAMHD1 (*Baldauf et al., 2012*). It has been recently demonstrated that HIV-1 latency was preferentially established in EMT CD4$^+$ T cells (*Shan et al., 2017*). Therefore, we investigated whether DFV-B infection in EMT CD4$^+$ T cells recapitulated the tendency of latency establishment. We first confirmed the primary CD4$^+$ T cells were in the process of EMT after 5–7 days in CFM culture, as the expression of CD25/CD69/HLA-DR and T cell exhaustion markers (PD-1 and TIM-3) had been greatly reduced (*Figure 3A and B*). mC cells expressed significantly lower levels of activation markers and exhaustion markers than DP cells (*Figure 3C and D*), again supporting our results that mC cells were latently infected. The infection profiles were then analyzed when infection was administered independently at different time points indicated (*Figure 3E*). In general, EMT CD4$^+$ T cells were less permissive to DFV-B infection; however, the ratio of mC cells to total infected cells (mC+DP) steadily increased from 15.7% (day 0 infection) to 64.8% (day 10 infection) (*Figure 3F*). This data strongly supports that infection during EMT process of CD4$^+$ T cells is most likely to generate latent infection, which also highlights the significance and applicability of our DFV-B system as a viable cell model for HIV-1 latency.

## Generation of a stable cell line model of HIV-1 latency

Although some cell line models of HIV-1 latency (such as the widely used J-Lat model) have been reported, all of which are derived from single-cell clones, and therefore with identical integration sites, which significantly underrepresented the heterogenous nature of latently infected cells in vivo. Here, by taking the advantage of direct labeling of latently infected cells with DFV-B infection, we generated a Jurkat-based cell line model of HIV-1 latency (J-mC), which were derived from random infection events, and therefore should harbor diverse integration sites. We found that DFV-B infection in Jurkat cells closely resembled primary CD4$^+$ T cells. Latently infected cells were readily identifiable on day 7 p.i. (*Figure 4—figure supplement 1A*). Then the DN, DP, and mC Jurkat T cells were sorted by flow cytometry and stimulated by 10 ng/ml TNF-α for 3 days in CFM. Over 80% of mC cells were converted to DP cells following the treatment, but DP and DN cells were not affected

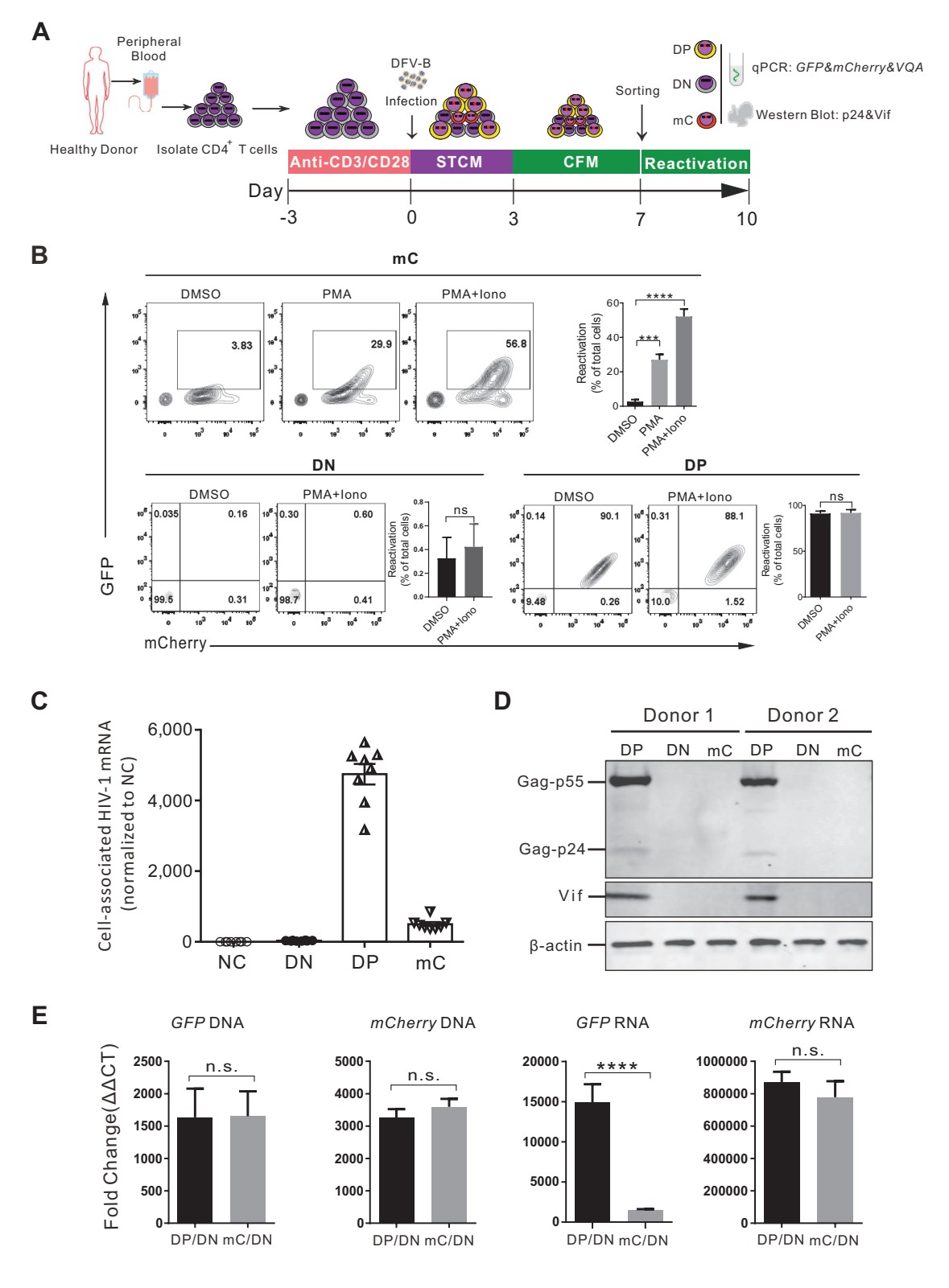

**Figure 2.** Validation of the DFV-B infected mC population as latently infected primary CD4+ T cells. (**A**) Schematic of experimental design. (**B**) Reactivation of the sorted different cell populations. Cells were cultured in CFM with 50 ng/ml PMA and 1 μM ionomycin and measured by flow cytometry 3 days later. Data shown from three different donors (mean ± SD). p-values calculated using unpaired t-test, \*\*\*p<0.001; \*\*\*\*p<0.0001; ns, not significant. (**C**) Quantification of cell-associated HIV-1 mRNA of different sorted populations. HIV-1 mRNAs were measured by real-time qPCR. RNA

*Figure 2 continued on next page*

*Figure 2 continued*
copies were normalized to the uninfected group. Each symbol represents the mean of three replicates (n = 8). (D) Western blot analysis of HIV-1 protein production in different sorted cell populations. (E) Quantification of the cell-associated DNA and RNA of GFP and mCherry in the sorted populations. The DNA and mRNA of GFP and mCherry were quantified relative to cellular GAPDH, respectively. Data shown from three biological replicates (mean ± SD). p-values calculated using unpaired t-test, ns, not significant; ****p<0.0001.
The online version of this article includes the following figure supplement(s) for figure 2:

**Figure supplement 1.** FACS sorting strategy and the viability of sorted cells.

(*Figure 4—figure supplement 1B*), confirming DFV-B infection could directly identify latent infection in Jurkat cells.

The process of model generation is illustrated in *Figure 4A*. Thirty days after the first sorting, a small dose of TNF-α (5 ng/ml) was added to the culture to activate those unstable cells that were probably in the borderline state of latency (*Figure 4A and B*). To assess the activatability of J-mC cells, we stimulated them with different stimuli. As expected, J-mC cells were activated efficiently by all stimuli to express both fluorescent proteins (*Figure 4C and D*). The potent induction of cell-associated HIV-1 mRNA after PMA treatment further confirmed the integrated proviruses were reactivated in this model (*Figure 4E*). Besides, mRNA expression levels and DNA levels of GFP and mCherry remained relatively stable over the course of a year (*Figure 4F and G*). Together, these results demonstrated that J-mC cells were indeed a stable group of latently infected cells and could be reactivated efficiently upon stimulation, suggesting that this would become a useful cell line model to study HIV-1 latency.

## The integration landscape of HIV-1 in J-mC cells

To verify whether J-mC cells harbored diverse HIV-1 integration sites, we adopted a method called linear amplification–mediated high-throughput genome-wide sequencing (LAM-HTGTS) (*Hu et al., 2016*) to analyze HIV-1 integration sites. We first tested the applicability of the method on J-Lat 8.4 and J-Lat 15.4, which had been reported to have only one HIV-1 integration site respectively (*Symons et al., 2017*). J-Lat 8.4 and 15.4 were mixed in equal amounts and then subjected to LAM-HTGTS. As shown in *Figure 5A and B*, 49.09% of the integration sites were located in the same position 77946384 of chromosome 1% and 46.92% counts were located in the same position 34441293 of chromosome 19 (*Supplementary file 1*), which were exactly the integration sites of J-Lat 8.4 and J-Lat 15.4, suggesting that LAM-HTGTS was a robust method for analyzing HIV-1 integration sites. For J-mC cells, all integration sites recovered were diffusely distributed on 24 chromosomes (*Figure 5C*). Chromosome 17 harbored the most abundant integration sites, which at the same time were extremely diverse (*Figure 5D* and *Supplementary file 2*). We also noticed that more than 70% of the integration sites were distributed on intronic regions (*Figure 5E*), which is consistent with previous reports on integration sites recovered from ART-treated HIV-1 infected individuals (*Han et al., 2004*; *Cohn et al., 2015*). We performed gene ontology (GO) and GO pathway analysis on the genes at all integration sites, and found that the proviruses were enriched in genes related to 'Regulation of transcription', 'Membrane', and 'Binding' (*Figure 5F*). In addition, HIV-1 integrated more frequently in cancer-related pathways (*Figure 5G*), which is also consistent with previously reported ex vivo data (*Wagner et al., 2014*; *Cohn et al., 2015*). Together, these data clearly demonstrated that J-mC cells harbored tremendously diverse HIV-1 integration sites, which were highly physiologically relevant, suggesting this cell line model as a valuable tool to study HIV-1 latency.

## High-throughput screening of small compound library in J-mC cells discovers ACY-1215 as a potent latency reversing agent

To test the applicability of our J-mC model, we performed a high-throughput screening of a small-molecule library containing 1700 FDA-approved compounds to discover novel LRAs. The screening was performed from high concentrations to low concentrations (*Figure 6—figure supplement 1A*), and with this system we successfully identified several previously reported LRAs, including SAHA, panobinostat, and romidepsin (*Figure 6A*). A new compound, ricolinistat (ACY-1215), was discovered in this screen (*Figure 6A and B*), displaying excellent potential to reactivate latent HIV-1 in a dose-dependent manner (*Figure 6C*, *Figure 6—figure supplement 1B*).

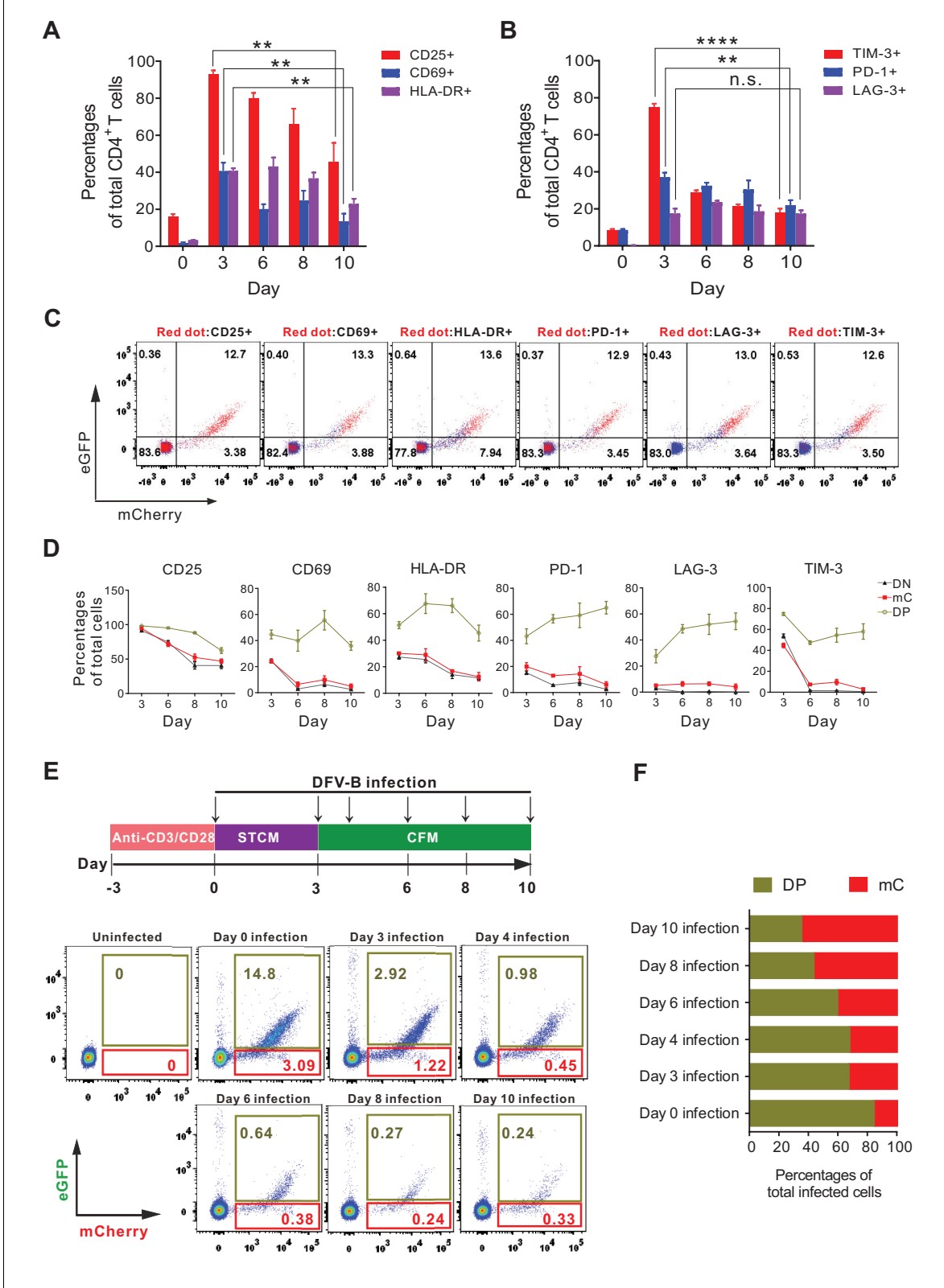

**Figure 3.** Effector-to-memory transitioning CD4+ T cells are the primary targets for HIV-1 latency establishment. (**A and B**) Expression of activation markers (CD25, CD69, HLA-DR; panel **A**) and exhaustion markers (LAG-3, TIM-3, PD-1; panel **B**) of the DFV-B infected primary CD4+ T cells. Primary CD4+ T cells were infected as described in *Figure 1B* and the surface markers were measured by FACS at indicated time points. Data indicate mean ± SD for three different donors. p-values calculated using unpaired t-test, ns, not significant; **p<0.01; ***p<0.001. (**C and D**) Expression of

*Figure 3 continued on next page*

*Figure 3 continued*

activation markers and exhaustion markers in the DN, DP, and mC population from **A** and **B** of this figure. Panel **C** only shows the results of day 8 post-infection. Red dots represented marker-positive cells; blue dots represented negative cells. Panel **D** shows quantified values of markers expression. Data indicate mean ± SD for three different donors. (**E**) Infection profile of EMT CD4$^+$ T cells when infected by DFV-B at indicated time points. Top panel depicts scheme of experimental procedure. Bottom panel shows the flow cytometric chart of infected EMT CD4$^+$ T cells on day 5 post-infection. (**F**) Infected EMT CD4$^+$ T cells are more likely to become latently infected cells. The ratios of DP cells or mC cells to total infected cells were calculated using data from panel **E**. Data represents the average of three donors.

Viral transcription was induced eight fold in average after 24 hr of ACY-1215 treatment (*Figure 6D*). In addition, we found that ACY-1215 showed better latency reversing activities in J-mC cells than in different J-Lat cell lines (*Figure 6—figure supplement 1C–E*), which probably explained why it was not discovered in previous drug screenings. Besides, we observed that ACY-1215 displayed a synergistic effect with several other LRAs, but not disulfiram (*Figure 6—figure supplement 1F*).

ACY-1215 is an oral HDAC6-selective inhibitor, and our data suggested that it was less toxic than SAHA as measured by annexin-V/propidium iodide (PI) staining (*Figure 6—figure supplement 2A and B*) and CCK-8 assay (*Figure 6—figure supplement 2C*). ACY-1215 does not induce upregulation of classic T cell activation markers on freshly isolated primary CD4$^+$ T cells (*Figure 6—figure supplement 3A*). In addition, ACY-1215 had no significant effect on CCR5 and CXCR4 expression of CD4$^+$ T cells (*Figure 6—figure supplement 3B and C*). No significant induction of IFN-γ and TNF-α was detected in CD8$^+$ T cells and CD4$^+$ T cells from the same donors following ACY-1215 treatment (*Figure 6—figure supplement 3D–G*). Taken together, these data indicated that ACY-1215 does not cause global T cell activation, T cell dysfunction, or cytotoxicity to CD4$^+$ and CD8$^+$ T cells.

To further verify the physiological relevance of the J-mC model, we tested the latency reversal activities of ACY-1215 in a primary cell model of HIV-1 latency and in primary cells from ART-treated infected individuals. Average 20 million of GFP$^-$ Bcl-2-transduced CD4$^+$ T cells were obtained from one healthy donor according to the protocol (*Kim et al., 2014*). At 48 hr post-treatment, we found that ACY-1215 worked as well as SAHA, reactivating over 80% of latent HIV-1 when comparing to PMA and ionomycin (*Figure 6E and F*). We collected five PBMC samples from ART-suppressed, HIV-1 infected individuals (*Figure 6—source data 1*) and isolated CD4$^+$ T cells. As shown in the *Figure 6G and H*, ACY-1215 robustly induced the level of cell-associated HIV-1 mRNA by an average of 50-fold comparing to the negative controls. Taken together, our data strongly supports that ACY-1215 is a promising LRA candidate and J-mC is a highly physiological relevant cell model for HIV-1 latency.

## Discussion

In this study, we presented a newly designed dual fluorescent virus, DFV-B. With DFV-B infection, latently infected cells can be easily distinguished from productively infected or uninfected cells. Compared with the previously reported infection models, DFV-B displayed a better infection profile with a clearer cell population and a lower percentage of single GFP positive cells (<1%) (*Figure 1B*) than RGH (*Dahabieh et al., 2013*), Duo-Fluo I (*Calvanese et al., 2013*), mdHIV (*Hashemi et al., 2016*), and HIV$_{GKO}$ (*Battivelli et al., 2018*).

More than 50% of the latently infected primary cells by DFV-B were reactivated by PMA and ionomycin in CFM (*Figure 2B*), while only 5% were reactivatable in the HIV$_{GKO}$ model (*Battivelli et al., 2018*), indicating the robustness of the DFV-B system to research the mechanism of HIV-1 latency maintenance and reactivation in vitro. A small fraction of single GFP positive cells existed after DFV-B infection (*Figure 1B*), but compared with Duo-Fluo I or HIV$_{GKO}$ infected cells, the proportion of the cells was extremely small (<1%). The eGFP and mCherry have 20–30 bp regions of homology at the C- and N-termini (*Salamango et al., 2013*), which may cause genomic deletion due to homologous recombination. However, our data showed that the absence of GFP production was due to viral latency but not genomic deletion (*Figure 2E*). Nevertheless, the risk of recombination still exists between GFP and mCherry, which may need further optimizations. Based on sequence similarity of fluorescent proteins that have been reported, E2 crimson/mKO2 may be a good choice to replace mCherry, which have been used previously to construct a dual-fluorescence reporter system

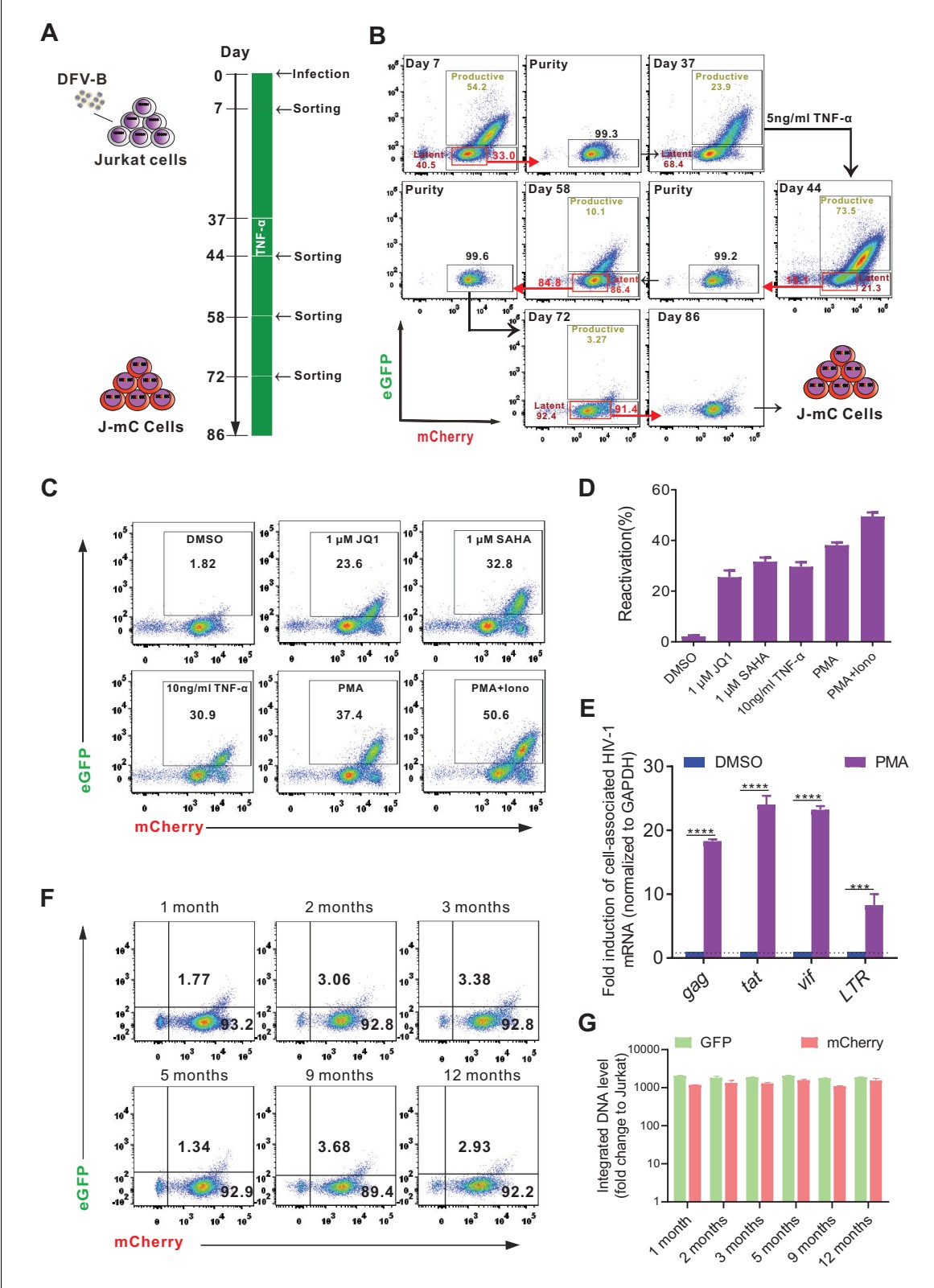

**Figure 4.** Generation of a novel cell line model of HIV-1 latency. (**A**) The experimental procedure of generating a Jurkat-based cell line model of HIV-1 latency (J-mC). Jurkat cells were infected by DFV-B at day 0 in CFM. (**B**) Flow cytometric charts during the process of J-mC model generation. Detailed steps can be found in 'Generation of J-mC cells' in the Materials and methods section. (**C**) Reactivation of J-mC cells by different stimuli. J-mC cells were stimulated, respectively, with 1 µM JQ1, 1 µM SAHA, 10 ng/ml TNF-α, 50 ng/ml PMA, and 50 ng/ml PMA and 1 µM ionomycin for 48 hr prior to

*Figure 4 continued on next page*

*Figure 4 continued*

analysis by flow cytometry. (**D**) Data from panel **C** are displayed graphically. Data indicate mean ± SD for three different treatments. (**E**) Cell-associated HIV-1 mRNA in J-mC cells after reactivation. J-mC cells were treated with DMSO or 50 ng/ml PMA for 48 hr. HIV-1 *gag*, *tat*, *vif*, and *LTR* mRNAs were quantified relative to cellular GAPDH, and values within each group were normalized to the DMSO treatment. Error bars represent SDs of three technical replicates. p-values calculated using unpaired t-test, ***p<0.001; ****p<0.0001. (**F**) Long-term tracking of the phenotype of J-mC cells. J-mC cells were cultured in CFM and the expression of GFP and mCherry were monitored over the course of a year. (**G**) Long-term tracking of the integrated DNA level of GFP and mCherry in J-mC cells. J-mC cells were collected on the date indicated, and then total DNA were isolated. The DNA of GFP and mCherry were quantified relative to cellular GAPDH, respectively. Data shown from three biological replicates (mean ± SD).

The online version of this article includes the following figure supplement(s) for figure 4:

**Figure supplement 1.** FACS sorting strategy and reactivation of the DP, DN, and mC Jurkat cells.

(*Battivelli et al., 2018*; *Kim et al., 2019*). Besides, the insertion location of the fluorescent protein in the construct needs to be considered, because there was a big difference in protein expression depending on whether 31 bp *nef* was retained or not, or two fluorescent proteins were connected or not in DFV-B.

How HIV-1 latency is established is still not fully understood. A widely accepted view is that after infection occurs in activated CD4[+] T cells, a tiny fraction of the infected cells survive viral cytopathic effect and host immune elimination and revert to a more resting and memory phenotype, becoming latently infected cells (*Sengupta and Siliciano, 2018*; *Margolis et al., 2020*). Our data here was in line with the mainstream view (*Figure 1B and C*). It is also suggested that latent infection is preferentially established in CD4[+] T cells undergoing EMT (*Chavez et al., 2015*; *Shan et al., 2017*). Our results from DFV-B infection indicated that EMT CD4[+] T cells were less permissive to infection, but once infected were much more likely to establish latency (*Figure 3E and F*), which supports that EMT CD4[+] T cells are the primary targets for latent infection, and highlights the feasibility of our DFV-B system.

Some primary cell models of HIV-1 latency have been developed in CD4[+] T cells or thymocytes (*Sahu et al., 2006*; *Burke et al., 2007*; *Marini et al., 2008*; *Bosque and Planelles, 2009*). However, the limited source and poor long-term viability of primary CD4[+] T cells preclude their extensive use. Yang et al reported a Bcl-2-transduced primary CD4[+] T cell model of HIV-1 latency, which could obtain a large amount of viable resting CD4[+] T cells long term and mimic the process of latency establishment in vitro. However, the over-expression of Bcl-2 may alter the global transcriptional profile and affect the physiological characteristics of the latently infected cells (*Yang et al., 2009*). In addition, the in vitro cell line model of latency, such as the J-Lat cell lines (*Jordan et al., 2003*), was derived from single-cell clone with identical HIV-1 integration sites, which significantly underrepresented the heterogeneous nature of latently infected cells (*Golumbeanu et al., 2018*; *Ait-Ammar et al., 2019*). Our novel model of J-mC cells harbored extremely diverse HIV-1 integration sites, with a majority of them located in intronic regions, which was in line with data recovered from infected individuals (*Han et al., 2004*, *Craigie and Bushman, 2012*, *Cohn et al., 2015*). In general, our DFV-B model system in primary CD4[+] T cells or Jurkat cells circumvented the problems mentioned above, and therefore could be utilized as a robust model to study the characteristics and mechanisms of HIV-1 latent infection.

By applying the J-mC model, we discovered a novel compound ricolinistat (ACY-1215) that can reactivate latent HIV-1 efficiently in different cell models and primary CD4[+] T cells from ART-treated infected individuals. ACY-1215, an orally bioavailable HDAC6-selective inhibitor, had previously completed Phase 1 and 2 clinical trials for treatment of multiple myeloma (*Santo et al., 2012*). In vitro, ACY-1215 selectively increased acetylated a-tubulin (a marker of HDAC6 inhibition) without altering acetylated histone H4 (a marker of global HDAC inhibition) at biologically relevant concentrations (<10 μM) (*Carew et al., 2019*). In osteoarthritis chondrocytes, the function of ACY-1215 was investigated at concentrations of 1–10 μM, and decreased cell viability was observed at a concentration of 50 μM (*Cheng et al., 2019*; *Li et al., 2019*). In the present study, we used 3 μM ACY-1215 in all additional assays to achieve selective inhibition of HDAC6 activity, which was in line with the studies mentioned above and was in the achievable range for clinical trials.

HDAC6 belongs to class II b HDACs and is a cytosolic microtubule-associated deacetylase. Selective HDAC6 inhibition stops aggresome formation, thence inhibiting the degradation leading to accumulation of misfolded proteins within cells, and may reduce the toxicity related to the off-target

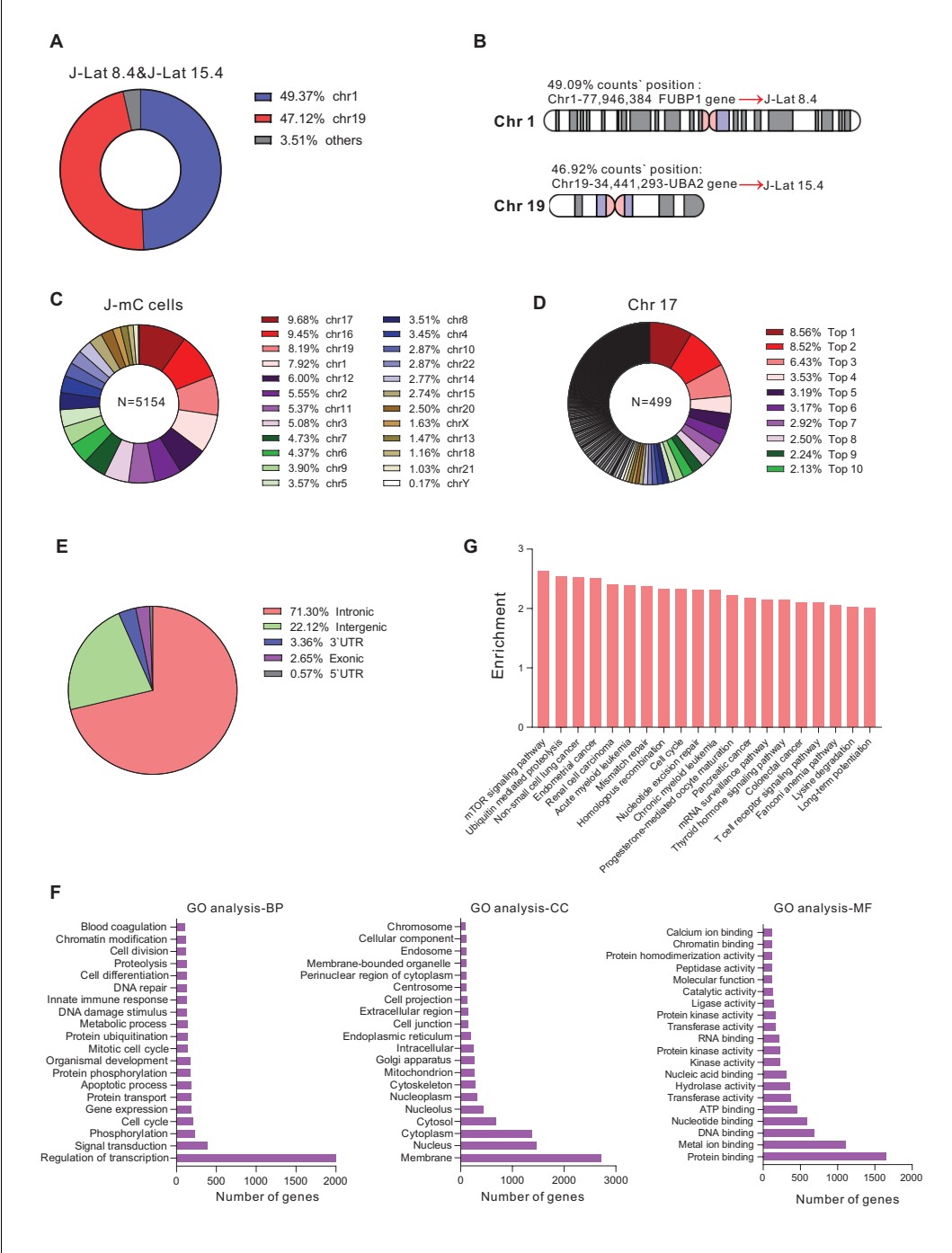

**Figure 5.** The integration landscape of HIV-1 in J-mC cells. (**A and B**) Validation of integration sites of J-Lat cell lines by LAM-HTGTS. J-Lat 8.4 and J-Lat 15.4 were mixed in equal amounts and subjected to LAM-HTGTS. **A**: Chromosome distribution of all integration sites. **B**: Integration sites analysis of J-Lat 8.4 and J-Lat 15.4. (**C and D**) Diversity of HIV-1 integration sites in J-mC cells. **C**: Chromosome distribution of all integration sites. A total of 5154 identified integration sites were obtained from J-mC cells. **D**: Distribution of integration sites on chromosome 17. The percentage of top 10 integration sites was shown. A total of 499 identified integration sites were obtained from chromosome 17. (**E**) Proportion of integration sites of J-mC cells in the indicated genomic regions. (**F**) The gene ontology analysis of HIV-1 integration sites detected in J-mC cells. (**G**) The GO pathway enrichment analysis of HIV-1 integration sites in J-mC cells.

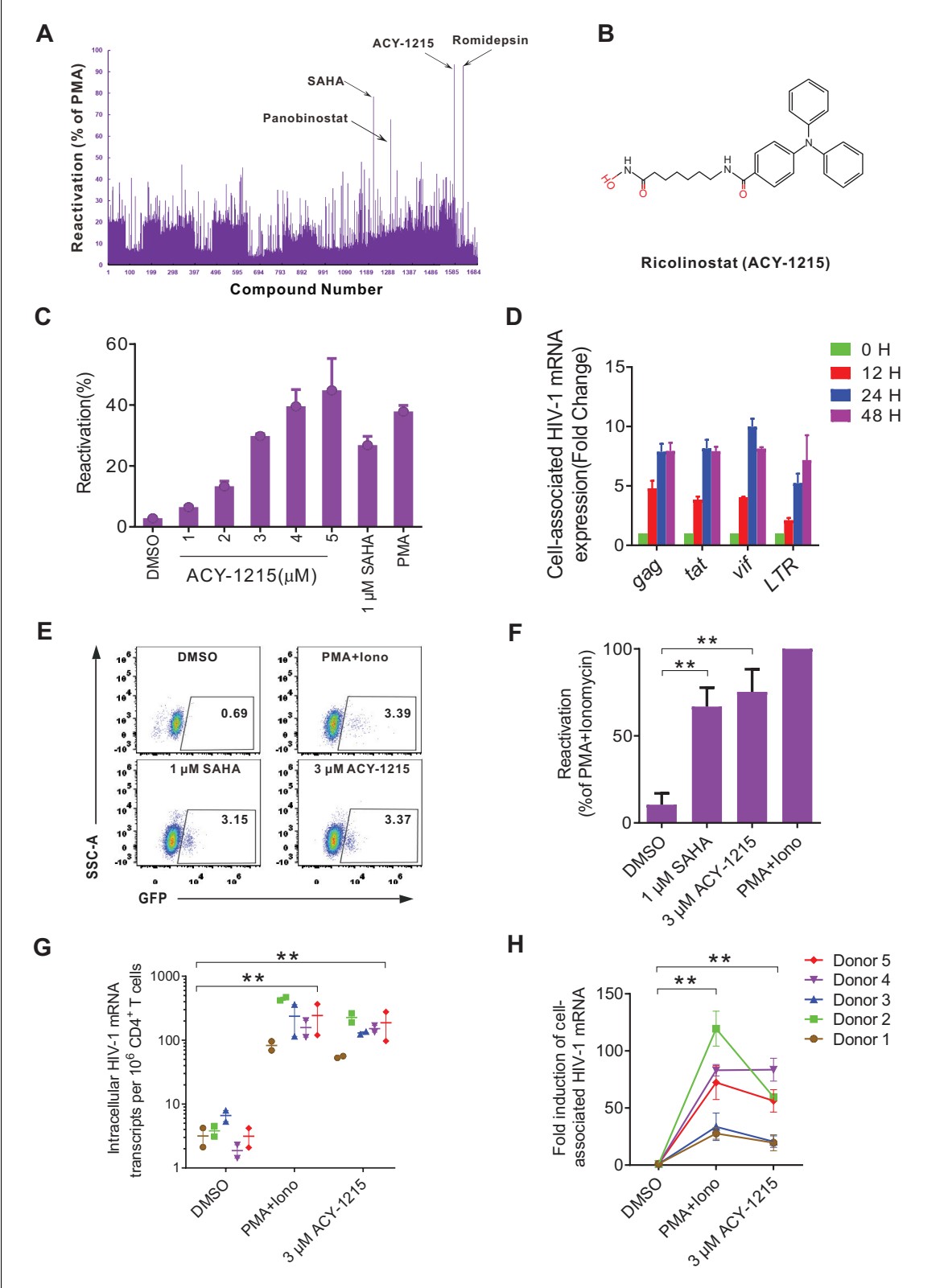

**Figure 6.** High-throughput screening with the J-mC model identifies ACY-1215 as a potent latency reversing agent. (**A**) Summary of screening results in J-mC cells. The results were shown as the percentage of the double positive cells and were normalized to the reactivation level of 50 ng/ml PMA. Each bar represents the average value of triplicates. (**B**) The chemical structure of ricolinostat (ACY-1215). (**C**) Reactivation effects of ACY-1215 on J-mC cells at the indicated concentrations. 0.5 million J-mC cells were stimulated with ACY-1215 for 3 days in each condition. Error bars represent SDs of three

*Figure 6 continued on next page*

*Figure 6 continued*

technical replicates. (D) Cell-associated HIV-1 mRNA expression of J-mC cells after ACY-1215 treatment. Data are representative of at least three independent experiments (mean ± SD). (E) ACY-1215 induced HIV-1 transcription in a primary CD4$^+$ T cells model of HIV-1 latency. (F) Data from panel E are displayed graphically. Histograms show quantification of the percent population in the active gate. Values represent the mean ± SD, N = 3. **p<0.01. Results were analyzed by unpaired t-tests. (G) Quantification of ACY-1215 induced transcription of cell-associated HIV-1 mRNA in CD4$^+$ T cells from ART-suppressed infected individuals. HIV-1 transcription was measured by HIV-1 viral quality assurance assay. 1 million CD4$^+$ T cells were analyzed per condition. Data indicate mean ± SD for three technical replicates. p-values calculated using unpaired t-test, **p<0.01. (H) Fold induction of cell-associated HIV-1 mRNA was shown relative to the negative control (DMSO). Data indicate mean ± SD for three technical replicates. p-values calculated using unpaired t-test, **p<0.01.

The online version of this article includes the following source data and figure supplement(s) for figure 6:

**Source data 1.** Characteristics of ART-suppressed, aviremic people living with HIV-1 (PLWH) in this study.
**Figure supplement 1.** Reactivation effects of ACY-1215 on J-mC cells and J-Lat cells.
**Figure supplement 2.** ACY-1215 did not cause apparent cytotoxicity.
**Figure supplement 3.** Effects of ACY-1215 on T cell biomarkers and function.

effects of pan-HDAC inhibitors (*Amengual et al., 2021*). Besides, HDAC6-KO mice were normal, as opposed to the lethal effect of genetic ablation of class I HDACs (*Haberland et al., 2009*). ACY-1215 and ACY-241 (another HDAC6-selective inhibitor) can downregulate Th2 cytokine production, decrease T-cell exhaustion, increase T-cell killing ability, and augment the proportion of central memory cells and expression of cytolytic markers in vitro (*Laino et al., 2019*; *Bae et al., 2018*). ACY-1215 can also change the chromatin accessibility of T cells (*Laino et al., 2019*). Besides, ACY-1215 has anti-inflammatory effect and is considered a potential anti-inflammatory drug (*Cheng et al., 2019*; *Zhang et al., 2018*; *Zhang et al., 2019*). A recent study reported that ACY-1215 could prevent the neurotoxicity of HIV-1 envelope protein gp120 (*Wenzel et al., 2019*). Our data suggested that ACY-1215 was less toxic than SAHA as described clinically (*Santo et al., 2012*), did not elevate the expression of classic T cell activation markers, and had no significant effect on HIV-1 co-receptors. Taken together, ACY-1215 could become a promising LRA for future AIDS functional cure studies.

In summary, our study designed and put forward a new dual fluorescent reporter virus, DFV-B, infection of which can directly mark latently infected primary CD4$^+$ T cells. We have also developed a stable latently infected cell line model based on DFV-B infection, which can be utilized as a valuable tool to study HIV-1 latency. We further identified a novel small compound ACY-1215 that can effectively reactivate latent HIV-1 and will be of great importance for the future applications of HIV-1 functional cure strategy.

# Materials and methods

## Key resources table

| Reagent type (species) or resource | Designation | Source or reference | Identifiers | Additional information |
|---|---|---|---|---|
| Antibody | Anti-human CD4-FITC (mouse monoclonal) | Biolegend | Cat# 317408; RRID:AB_571951 | Dilution 1:1000 |
| Antibody | Anti-human CD8-APC (mouse monoclonal) | Biolegend | Cat# 301049; RRID:AB_2562054 | Dilution 1:1000 |
| Antibody | Anti-human Bcl2-AF647 (mouse monoclonal) | Biolegend | Cat# 658706; RRID:AB_2563280 | Dilution 1:1000 |
| Antibody | Anti-human CD69-APC (mouse monoclonal) | Biolegend | Cat# 310910; RRID:AB_314845 | Dilution 1:1000 |
| Antibody | Anti-human CD25-APC (mouse monoclonal) | Biolegend | Cat# 302610; RRID:AB_314280 | Dilution 1:1000 |
| Antibody | Anti-human HLA-DR-APC (mouse monoclonal) | Biolegend | Cat# 307610; RRID:AB_314688 | Dilution 1:1000 |

*Continued on next page*

*Continued*

| Reagent type (species) or resource | Designation | Source or reference | Identifiers | Additional information |
|---|---|---|---|---|
| Antibody | Anti-human PD-1-APC (mouse monoclonal) | Biolegend | Cat# 329908; RRID:AB_940475 | Dilution 1:1000 |
| Antibody | Anti-human LAG-3-APC (mouse monoclonal) | Biolegend | Cat# 369212; RRID:AB_2728373 | Dilution 1:1000 |
| Antibody | Anti-human CCR5-APC (mouse monoclonal) | Biolegend | Cat# 359122; RRID:AB_2564073 | Dilution 1:1000 |
| Antibody | Anti-human CXCR4-APC (mouse monoclonal) | Biolegend | Cat# 306510; RRID:AB_314616 | Dilution 1:1000 |
| Antibody | Anti-human IFN-γ-APC (mouse monoclonal) | Biolegend | Cat# 502512; RRID:AB_315237 | Dilution 1:1000 |
| Antibody | Anti-human TNF-α-APC (mouse monoclonal) | Biolegend | Cat# 502912; RRID:AB_315264 | Dilution 1:1000 |
| Antibody | Anti-HIV-1 p24 antibody (mouse monoclonal) | ABcam | Cat# ab9071; RRID:AB_306981 | Dilution 1:1000 |
| Antibody | Anti-HIV-1 Vif antibody (mouse monoclonal) | ABcam | Cat# ab66643; RRID:AB_1139534 | Dilution 1:1000 |
| Antibody | Anti-β-actin antibody (mouse monoclonal) | ABcam | Cat# ab8226; RRID:AB_306371 | Dilution 1:1000 |
| Antibody | Goat Anti-Mouse IgG H and L (Alexa Fluor 680) preadsorbed | ABcam | Cat# ab186694 | Dilution 1:20000 |
| Antibody | Purified anti-human CD3 Antibody (mouse monoclonal) | Biolegend | Cat# 300302; RRID:AB_314038 | Dilution 1:1000 |
| Antibody | Purified anti-human CD28 Antibody (mouse monoclonal) | Biolegend | Cat# 302902; RRID:AB_314304 | Dilution 1:1000 |
| Strain, strain background (*Escherichia coli*) | Stbl3 *E. coli* | ThermoFisher | Cat#C7381201 | |
| Biological sample (*Homo sapiens*) | Blood samples | Guangzhou Blood Center, Guangzhou | http://www.gzbc.org/ | Blood samples from healthy individuals |
| Biological sample (*Homo sapiens*) | Blood samples | The Fifth Affiliated Hospital, Sun Yat-sen University, Zhuhai, China | http://www.zsufivehos.com/ | Blood samples from HIV-1- infected individuals |
| Tissue culture media | RPMI 1640 | GIBCO | Cat# 11875093 | |
| Tissue culture media | DMEM | GIBCO | Cat# 11995065 | |
| Tissue culture media | Penicillin-Streptomycin Solution | BBI Life Sciences | Cat# E607011 | |
| Tissue culture media | 1 M Hepes Solution | BBI Life Sciences | Cat# E607018 | |
| Chemical compound, drug | DMSO | Sigma-Aldrich | Cat# D2650-100ML | |
| Chemical compound, drug | Disulfiram | Selleckchem | Cat# S1680 | |
| Chemical compound, drug | Bryostatin-1 | Sigma-Aldrich | Cat# B7431 | |
| Chemical compound, drug | Panobinostat | Selleckchem | Cat# S1030 | |

*Continued on next page*

*Continued*

| Reagent type (species) or resource | Designation | Source or reference | Identifiers | Additional information |
|---|---|---|---|---|
| Chemical compound, drug | ACY-1215 | Selleckchem | Cat# S8001 | |
| Chemical compound, drug | (+)-JQ-1 | Selleckchem | Cat#S7110 | |
| Chemical compound, drug | SAHA | Selleckchem | Cat#S1047 | |
| Chemical compound, drug | Approved Drug Library | TargetMOI | Cat# L1000 | |
| Chemical compound, drug | Phorbol 12-myristate 13-acetate (PMA) | Selleckchem | Cat#S7791 | |
| Chemical compound, drug | Ionomycin | MERCK | at#407952–5 MG | |
| Chemical compound, drug | Phytohemagglutinin -M(PHA-M) | Sigma-Aldrich | Cat#11082132001 | |
| Chemical compound, drug | TRIzol Reagent | ThermoFisher | Cat#15596018 | |
| Chemical compound, drug | Propidium Iodide (PI) | Biolegend | Cat# 421301 | |
| Chemical compound, drug | Annexin-V-APC | Biolegend | Cat# 640941; RRID:AB_2616657 | |
| Recombinant proteins | Recombinant Human TNF-a | PeproTech | Cat#300-01A | |
| Recombinant proteins | Recombinant Human IL-2 | R and D Systems | Cat#202-IL-500 | |
| Critical commercial assays | Human CD4+ T Lymphocyte Enrichment Set-DM | BD Biosciences | Cat#557939 | |
| Critical commercial assays | HIV-1 p24 ELISA Kit | Abcam | Cat#ab218268 | |
| Critical commercial assays | Cell Counting Kit-8 | MedChemExpress | Cat# HY-K0301 | |
| Critical commercial assays | Plasmid Mini Kit | OMEGA | Cat# D6943-02 | |
| Critical commercial assays | Endo-free Plasmid Mini Kit | OMEGA | Cat# D6950-02 | |
| Critical commercial assays | Gel Extraction | OMEGA | Cat# D2500-02 | |
| Critical commercial assays | Cycle-Pure Kit | OMEGA | Cat# D6492-02 | |
| Critical commercial assays | Tissue DNA kit | OMEGA | Cat# D3396-02 | |
| Cell line (*Homo sapiens*) | HEK293T | ATCC | CRL-3216; RRID:CVCL_0063 | |
| Cell line (*Homo sapiens*) | J-Lat 6.3 | NIH AIDS Reagents Program (*Jordan et al., 2003*) | Cat# 9846; RRID:CVCL_8280 | |

*Continued on next page*

*Continued*

| Reagent type (species) or resource | Designation | Source or reference | Identifiers | Additional information |
|---|---|---|---|---|
| Cell line (*Homo sapiens*) | J-Lat 8.4 | NIH AIDS Reagents Program (*Jordan et al., 2003*) | Cat#9847; RRID:CVCL_8284 | |
| Cell line (*Homo sapiens*) | J-Lat 9.2 | NIH AIDS Reagents Program (*Jordan et al., 2003*) | Cat# 9848; RRID:CVCL_8285 | |
| Cell line (*Homo sapiens*) | J-Lat 10.6 | NIH AIDS Reagents Program (*Jordan et al., 2003*) | Cat#9849; RRID:CVCL_8281 | |
| Cell line (*Homo sapiens*) | J-Lat 15.4 | NIH AIDS Reagents Program (*Jordan et al., 2003*) | Cat# 9850; RRID:CVCL_8282 | |
| Cell line (*Homo sapiens*) | J-Lat A2 | NIH AIDS Reagents Program (*Jordan et al., 2003*) | Cat#9867; RRID:CVCL_1G43 | |
| Software | Prism 6 | GraphPad | https://www.graphpad.com/scientific-software/prism/; RRID:SCR_002798 | |
| Software | BD LSRFortessa cell analyzer | BD Biosciences | http://www.bdbiosciences.com/in/instruments/lsr/index.jsp; RRID:SCR_018655 | |
| Software | FlowJo V10 | Tree Star | https://www.flowjo.com/; RRID:SCR_008520 | |
| Software | GloMax 96 Microplate Luminometer Software | Promega | https://www.promega.com/resources/softwarefirmware/; RRID:SCR_018614 | |
| Sequence-based reagent | HIV-1-VQA-F | This paper | PCR primers | CAGATGCTGCATATAAGCAGCTG |
| Sequence-based reagent | HIV-1-VQA-R | This paper | PCR primers | TTTTTTTTTTTTTTTTTTTTTTTTGAAGCAC |
| Sequence-based reagent | HIV-1-VQA-Probe | This paper | PCR primers | FAM-CCTGTACTGGGTCTCTCTGG-MGB |
| Sequence-based reagent | qPCR-GFP-F | This paper | PCR primers | GTGCAGTGCTTCAGCCGCTACC |
| Sequence-based reagent | qPCR-GFP-R | This paper | PCR primers | ACCTCGGCGCGGGTCTTGTA |
| Sequence-based reagent | qPCR-mCherry-F | This paper | PCR primers | GTGGTGACCGTGACCCAGGACT |
| Sequence-based reagent | qPCR-mCherry-R | This paper | PCR primers | TGGTCTTGACCTCAGCGTCGTAGTG |
| Sequence-based reagent | qPCR-GAPDH-F | This paper | PCR primers | CTCTGCTCCTCCTGTTCGAC |
| Sequence-based reagent | qPCR-GAPDH-R | This paper | PCR primers | AGTTAAAAGCAGCCCTGGTGA |
| Sequence-based reagent | HIV-1-gag-F | This paper | PCR primers | ACATCAAGCAGCCATGCAAAT |
| Sequence-based reagent | HIV-1-gag-R | This paper | PCR primers | TCTGGCCTGGTGCAATAGG |
| Sequence-based reagent | HIV-1-tat-F | This paper | PCR primers | ATGGAGCCAGTAGATCCTAGAC |
| Sequence-based reagent | HIV-1-tat-R | This paper | PCR primers | CGCTTCTTCCTGCCATAGG |
| Sequence-based reagent | HIV-1-vif-F | This paper | PCR primers | CACACAAGTAGACCCTGACCT |

*Continued on next page*

*Continued*

| Reagent type (species) or resource | Designation | Source or reference | Identifiers | Additional information |
|---|---|---|---|---|
| Sequence-based reagent | HIV-1-vif-R | This paper | PCR primers | CCCTACCTTGTTATGTCCTGCT |
| Sequence-based reagent | HIV-1-LTR-F | This paper | PCR primers | CCACAAAGGGAGCCATACAATG |
| Sequence-based reagent | HIV-1-LTR-R | This paper | PCR primers | TTATGGCTTCCACTCCTGCC |

## Cell lines

HEK293T and Jurkat cells were obtained from ATCC. HEK293T cells were cultured in DMEM supplemented with 1% penicillin–streptomycin (ThermoFisher), 1% L-glutamine (ThermoFisher), and 10% FBS (ThermoFisher). Jurkat T cells were cultured in CFM (RPMI 1640 supplemented with 1% penicillin–streptomycin, 1% L-glutamine, and 10% FBS). J-Lat 6.3, J-Lat 8.4, J-Lat 9.3, J-Lat 10.6, J-Lat 15.4, and J-Lat A2 cell lines were derived from the human Jurkat T cell line and were gifts from Dr. Robert F. Siliciano (Department of Medicine, Johns Hopkins University School of Medicine, Baltimore, MD) Laboratory, which were originally generated from Dr. Eric Verdin (The Buck Institute for Research on Aging, Novato, CA). All the J-Lat cell lines were cultured in CFM. All cells have been tested for mycoplasma using a PCR assay and confirmed to be mycoplasma-free. All cells were cultured in a sterile incubator at 37°C and 5% $CO_2$.

## Plasmids and vector construction

DFV-B was derived from the original NL4-3 strain of HIV-1, which contains an HIV-1 genome. To construct DFV-B, GFP (Fragment 2) and EF-1α-mCherry (Fragment 3) were obtained by cloning in the respective plasmids. A 347 bp sequence (Fragment 1) was amplified from pNL4-3 plasmid using primers F1: 5'-ATTAGTGAACGGATCCTTAGCACTTATCTGGGACG-3' and R1: 5'-GCCCTTGCTCACCATCTTATAGCAAAATCCTTTCCA-3'. A 3'LTR sequence (Fragment 4) was amplified from pNL4-3 plasmid using primers F2: 5'-CTTAGCCACTTTTTAAAAGAAA-3' and R2: 5'- ACCCTGCACTCCATGGATCAGCTGGCACCCCCGGA-3'. Next, a skeleton sequence was obtained from pNL4-3 using the *BamH* I and *Nco* I restriction sites. Finally, Fragment 1, GFP (Fragment 2), EF-1α-mCherry (Fragment 3), and Fragment 4 were cloned into the skeleton sequence by a modified SLIC technique (Li and Elledge, 2012). Of note, a portion of *env* was deleted for a single round of infection of the virus, and the *vpr* gene had a frameshift mutation to reduce the toxicity of the pseudovirus. Most of the *nef* sequence was replaced by GFP- EF-1α-kozak-mCherry, and the remaining 31 bp of nef sequence was reserved in order not to disturb the function of LTR.

## Virus stock production

Pseudotyped viral stocks were produced in HEK293 T cells by co-transfecting 2.2 µg of VSV-G glycoprotein-expression or pCXCR4 HIV-1 envelope-expression vector, 4.4 µg of packaging vector pC-Help, and 4.4 µg pEB-FLV or pDFV-B construct using a polyethylenimine (Invitrogen) transfection system according to the manufacturer's instructions. Supernatants were harvested after 48 hr, centrifuged (10 min, 500 × g, room temperature), and filtered through a 0.45 µm pore size membrane to remove the cell debris. Viruses were concentrated by centrifuging with a 25% vol of 50% polyethylene glycol 6000 and a 10% vol of 4 M NaCl. Concentrated virions were resuspended in complete medium and stored at −80°C.

## Generation of J-mC cells

The cell line model of HIV-1 latency was generated as described (*Figure 4A and B*). Briefly, Jurkat cells were infected by DFV-B incorporating a VSVG envelope at 250 ng virus per million cells on day 0 in CFM. At day 7, the single mCherry positive cells were sorted according to the red gate in *Figure 4B*. The purity of single mCherry positive cells was greater than 99%. Next, the cells were cultured for another 30 days in CFM. At day 37, a small dose of TNF-α (5 ng/ml) was added to the culture and maintained for 7 days to further activate those unstable cells that were probably in the

borderline state of latency. At day 44 the second sorting of single mCherry positive cells was performed by the red gate. Similarly, the third and fourth sorting of single mCherry positive cells were performed at days 58 and 72. J-mC cells were finalized at day 86. Each sorting was performed by the red gate in *Figure 4B*.

## Generation of latently HIV-1 infected Bcl-2–transduced cells

This primary cell of HIV-1 latency was generated as described previously (*Kim et al., 2014*). In brief, primary CD4[+] T cells were isolated from PBMCs with Human CD4[+] T lymphocyte Enrichment Set-DM (BD, USA) according to manufacturer's instructions. CD4[+] T cells were co-stimulated with anti-CD3 and anti-CD28 antibodies and were transduced with EB-FLV at the MOI of 5–10 by spinoculation of cells at 1200 g at room temperature for 2 hr. Following 3–4 weeks of culture, viable cells were isolated using Ficoll density gradient centrifugation. Activated Bcl-2-transduced cells were then infected with reporter virus NL4-3-Δ6-drEGFP at a MOI of less than 0.1. The infected cells were maintained in IL-2 and T cell growth factor–enriched medium for 3 days after infection, and then the medium was changed to RPMI 1640 with 10% FBS and 1% penicillin/streptomycin without exogenous cytokines and cultured for more than 1 month. Finally, the GFP-negative cells were sorted to more than 99.9% purity using fluorescence-activated cell sorting (FACS Aria II, BD).

## Infection of primary CD4[+] T cells

Primary CD4[+] T cells were isolated from HIV naïve individuals. CD4[+] T cells were co-stimulated with anti-CD3 and anti-CD28 antibodies for 3 days. Activated CD4[+] T cells were then infected with DFV-B incorporating a CXCR4-tropic Env at a MOI of 0.05–0.1. The infected cells were maintained in STCM for 3 days after infection, then the medium was changed to CFM, and cultured for 1 week.

## Flow cytometry and cell sorting

GFP and mCherry fluorescence were measured in a BD LSR Fortessa cell analyzer and sorted in a FACS Aria II (BD). Data were analyzed using Flowjo (version 10.4.0).

## DNA, RNA extraction, and qPCR

Total DNA was extracted with tissue DNA kit (OMEGA). Cells were incubated with different drugs for different time intervals, total RNA was extracted using TRIzol (Invitrogen) and chloroform, and then precipitated with isopropyl alcohol. RNA reverse transcription to cDNA was done according to the procedure of the SuperScript IV Control Reactions (ThermoFisher). Relatively quantitative PCR was performed using SYBR qPCR Master Mix (Vazyme) on the QuantStudio 3 (Thermo Fisher Scientific) using a standard two-step procedure (denatured: 95°C/10 s, annealed/extended: 60°C/30 s, 40 cycles). The RT-PCR primer sequences used were shown in Key resources table. The $2^{-\Delta\Delta CT}$ method was adopted to analyze the relative expression of mRNAs, using GAPDH as the reference gene.

## Viral quality assurance

Total cellular RNA was extracted from 1 million CD4[+] T cells after 24 hr of stimulation followed by reverse transcription. Real-time PCR was performed using the TaqMan (Life Technologies) gene expression assay. HIV-1 mRNAs were detected using the following primers and probe, modified from *Shan et al., 2013*. Forward (5′→3′) CAGATGCTGCATATAAGCAGCTG (9501–9523), Reverse (5′→3′) TTTTTTTTTTTTTTTTTTTTTTTTGAAGCAC (9629-poly A), and Probe (5′→3′) FAM-CCTGTACTGGGTCTCTCTGG-MGB (9531–9550). The cycling parameters were as follows: 2 min at 50°C, 10 min at 95°C, and 45–50 cycles at 95°C for 15 s and then 60°C for 60 s. Molecular standard curves were generated using serial dilutions of a TOPO plasmid containing the last 352 nucleotides of viral genomic RNA plus 30 deoxyadenosines (pVQA). Results for each drug treatment were presented as fold change relative to DMSO control (mean ± SD).

## Protein extraction and western blot

After treatment, cells were lysed in RIPA lysis buffer (10 mM Tris-HCl buffered at pH 7.5, 150 mM NaCl, 0.5 % NP-40, 1% Triton X-100, 10% glycerol, 2 mM EDTA, 1 mM NaF, 1 Mm $Na_3VO_4$, 1× protease and phosphatase inhibitor cocktail) and incubated in ice for 30 min, followed by centrifugation at 12,000 g for 10 min at 4°C. The supernatants were collected as a whole protein extract.

Protein samples were stored at −80°C or directly used for western blotting analysis. The protein extract was denatured by addition of NuPAGE running buffer (ThermoFisher), followed by denaturation at 100°C for 10 min, 50 µg of total protein were electrophoresis for 1.5 hr on a 10% polyacrylamide gel to separate the proteins, transferred onto PVDF membranes (Biorad), and co-incubation with primary and secondary antibodies. The antibodies were diluted by 5% BSA. Subsequently, Blots were imaged on an infrared (IR) laser-based fluorescence imaging system (LI-COR Odyssey).

### Linear amplification–mediated high-throughput genome-wide sequencing (LAM-HTGTS)

Genomic DNA is sheared by sonication, and the HIV-1-human genome junctions are then amplified by LAM-PCR (*Schmidt et al., 2007*), using directional primers lying on HIV-1 LTR. LAM-PCR with a single 5′ biotinylated primer amplifies across the HIV-1 LTR into the unknown human genome. Junction-containing single-stranded DNAs (ssDNAs) are enriched via binding to streptavidin-coated magnetic beads. After washing, bead-bound ssDNAs are unidirectionally ligated to a bridge adapter. Adapter-ligated, bead-bound ssDNA fragments are then subjected to nested PCR to incorporate a barcode sequence that is necessary for demultiplexing. A final PCR step fully reconstructs Illumina Miseq adapter sequences at the ends of the amplified HIV-1-human junction sequence. Samples are then separated on an agarose gel, and the resulting population of 0.5–1 kb fragments is collected and quantified before Miseq paired-end sequencing, with a typical 2 × 250 bp HTGTS library. The sequence of the matched read pairs after the LTR or adapter were trimmed to 50 base pairs and subsequently the trimmed read pairs were grouped according to 100% sequence match. Well-represented sequences were used for chromosomal alignment which was determined using the Blat-UCSC Genome Browser (GRCH38/hg38). An HIV integration site was identified if the position had ≥10 counts.

### Drug screening

We screened a library of 1700 drugs from the FDA-approved drug library. $1 \times 10^5$ J-mC cells were resuspended in 200 µl of medium in round-bottom 96-well plates. For the first round of screening, J-mC cells were treated with compounds (10 µM) for 48 hr and were analyzed by FACS. Then drugs with less than 30% of live cells are selected from the result of the first round of screening for a second round of screening with 5 µM drug concentration. Each plate contained cells treated with 50 ng/ml PMA, PBS and DMSO as controls. After 48 hr at 37°C, reactivation of latent HIV-1 was detected as above.

### Cytotoxicity assays

Cell death radio was evaluated by Annexin V/propidium iodide (PI) double staining assay. The cells incubate Annexin V-FITC at room temperature for 15 min. Then add PI before analyzing the stained cells by flow cytometry. Cell viability was measured by a Cell Count Kit-8 (MCE). PBMCs plated as 1 million cells per well in a 96-well plate were incubated with drugs for 48 hr. Then 10 µl of CCK-8 reagent was added to 100 µl of cell culture mixture and incubated for an additional 4 hr. Optical density (OD) was recorded at a wavelength of 450 nm on a microplate reader (Molecular Devices). The cell viability was calculated as follows. (OD treatment – OD blank control) / (OD untreatment – OD blank control) × 100%.

### Statistical analysis

All experiments were repeated at least three times. Statistics were performed with GraphPad Prism 8 (GraphPad), and all data are presented as mean values ± SD. p-values were calculated based on the Mann–Whitney U-test or Student's t-test (large sample size) using SPSS software. * denotes $p < 0.05$, ** denotes $p < 0.01$, *** denotes $p < 0.001$, **** denotes $p < 0.0001$, and NS denotes non-significant.

## Acknowledgements

We thank all the volunteers who donated blood samples for this study. This work was supported by National Natural Science Foundation of China (32070159 and 81672024), the National Special

Research Program of China for Important Infectious Diseases (2018ZX10302103 and 2017ZX10202102), Natural Science Foundation of Guangdong Province of China (2017A030306005), and Guangdong Innovative and Entrepreneurial Research Team Program (2016ZT06S638) to KD.

## Additional information

### Funding

| Funder | Grant reference number | Author |
| --- | --- | --- |
| National Natural Science Foundation of China | 81672024 | Kai Deng |
| National Basic Research Program of China | 2018ZX10302103 | Kai Deng |
| National Basic Research Program of China | 2017ZX10202102 | Kai Deng |
| National Natural Science Foundation of China | 32070159 | Kai Deng |
| Natural Science Foundation of Guangdong Province | 2017A030306005 | Kai Deng |
| Guangdong Innovative and Entrepreneurial Research Team Program | 2016ZT06S638 | Kai Deng |

The funders had no role in study design, data collection and interpretation, or the decision to submit the work for publication.

### Author contributions

Jinfeng Cai, Data curation, Formal analysis, Validation, Investigation, Methodology, Writing - original draft, Writing - review and editing; Hongbo Gao, Data curation, Methodology; Jiacong Zhao, Shujing Hu, Xinyu Liang, Yanyan Yang, Zhuanglin Dai, Data curation, Validation, Investigation; Zhongsi Hong, Resources, Investigation; Kai Deng, Conceptualization, Resources, Supervision, Funding acquisition, Methodology, Writing - original draft, Project administration, Writing - review and editing

### Author ORCIDs

Jinfeng Cai (iD) https://orcid.org/0000-0001-8356-030X
Kai Deng (iD) https://orcid.org/0000-0002-9973-8130

### Ethics

Human subjects: The use of PBMCs from healthy individuals was approved by the Institutional Review Board of Guangzhou Blood Center (Guangzhou, Guangdong, China). We did not have any interaction with the healthy individuals or protected information, and therefore no informed consent was required. Chronically HIV-1-infected participants sampled by this study were recruited from The Fifth Affiliated Hospital of Sun Yat-sen University (Zhuhai, Guangdong, China). This study was approved by the Ethics Review Boards of the Fifth Affiliated Hospital of Sun Yat-sen University (2018K41-1). All the participants were given written informed consent with approval of the Ethics Committees. The enrollment of HIV-1-infected individuals was based on the criteria of prolonged suppression of plasma HIV-1 viremia on cART, which is undetectable plasma HIV-1 RNA levels (less than 50 copies/ml) for a minimum of 12 months, and having high CD4$^+$ T cell count (at least 350 cells/mm$^3$). All HIV-1-infected participants provided written informed consent for their participation in the study and agreed with the publication of the scientific results.

### Decision letter and Author response

Decision letter https://doi.org/10.7554/eLife.63810.sa1

Author response https://doi.org/10.7554/eLife.63810.sa2

## Additional files

### Supplementary files
- Supplementary file 1. HIV-1 integration sites of J-Lat 8.4 and J-Lat 15.4.
- Supplementary file 2. HIV-1 integration sites of J-mC cells.
- Transparent reporting form

### Data availability
All data generated or analysed during this study are included in the manuscript.

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
