## [Decision Letter]

**Acceptance summary:**

The authors demonstrate that their dual reporter virus is an efficient tool to determine latency and to test latency reversal agents. These findings will contribute to the armoury to study and combat HIV latency.

**Decision letter after peer review:**

Thank you for submitting your article "Infection with a newly-designed dual fluorescent reporter HIV-1 effectively identifies latently infected CD4^+^ T cells" for consideration by *eLife*. Your article has been reviewed by 3 peer reviewers, one of whom is a member of our Board of Reviewing Editors, and the evaluation has been overseen by Miles Davenport as the Senior Editor. The following individual involved in review of your submission has agreed to reveal their identity: Thomas Rasmussen (Reviewer #2).

The reviewers have discussed the reviews with one another and the Reviewing Editor has drafted this decision to help you prepare a revised submission.

Summary:

In this paper, Cai and colleagues describe a dual fluorescent reporter virus, DFV-B, that was used to infect primary CD4^+^ T cells and Jurkat cells. In DFV-B infected primary CD4^+^ T cells, latently infected cells could be sorted out and in those more than 50% could be activated to produce GFP (under control of LTR) using PMA/ionomycin stimulation. In Jurkat cells the authors demonstrated the establishment of a latently infected cell line model of HIV latency that was used to identify ACY-1215, a selective inhibitor of HDAC6, as a potential latency-reversing agent.

The manuscript is well written and in general, data are clearly and soundly presented. As such, this work should be of interest for the field of HIV latency and cure, but there are a few outstanding issues that may require further clarification or modification:

Essential revisions:

1. The results show that the ratio of mC cells tot total infected cells increased over prolonged culturing in CD4 T cells. However, as the mC signal also dropped 10 fold. Additionally, there seems to be cells that solely transcribe the productive reporter.

The drop in the latent reporter needs to be explained. My understanding is that this reporter should be continuously transcribed. A loss of signal of the latent reporter is a known problem with dual reporter viral systems. Why does this system show less efficiency in primary cells compared to cell lines.

2. Figure 2: I suggest including the PMA/ionomycin stimulation of DN and DP cells in the actual Figure 2 (currently in the supplement) as these are good controls and add further credibility to the conclusion.

3. Figure 4: The process of model generation is not quite clear from text and Figure 4. Please further clarify identification of latently infected (mCherry) and DP cells. In that process I recommend incorporating the gating plots from the supplementary figure to further clarify gating strategy and how this is different to the sorting plots shown in Figure 4B. It is also not clear at which time points in the model (shown in Figure 4B) cells were sorted and then subjected to the various stimuli.

4. The authors mention that previous J-Lat models are disadvantaged due to their identical integration sites, but that their J-mC model of DFB-V infection should harbour diverse integration sites. Yet, no data are presented to confirm this assumption.

5. The use of the developed model systems in the identification of ACY-1215 is

interesting and relevant, but several important pieces of information are missing, which will further strengthen the presentation of these data:

a. The dose-dependent activation of GFP expression with increasing concentration of ricolinistat looks consistent and credible, but information on how long cells were stimulated for should be included in the figure or figure text, as should the utilised concentration of SAHA. It would also strengthen the interpretation if the authors could relate the used concentrations of ricolinistat to therapeutic levels during clinical dosing, ie are these levels physiologically relevant?

b. Figure 5G: Given the significant variation among individuals in cell-associated HIV RNA in the absence of stimulation, it would useful to also display the actual measured levels across DMSO, PMA/ionomycin and ACY conditions in Figure 5G. Also, which concentration of ACY was used here and is this therapeutically relevant? Finally, how many isolated CD4^+^ T cells were analysed per condition?

6. Depending on the above clarifications with regard to the most relevant concentration of ricolinistat for ex vivo and in vitro use, the compound looks like it could have similar potency to vorinostat, which is not overly impressive and may not offer any advantage compared to previously tested LRAs. One difference could be its HDAC6 selectivity, which may offer a better effect/toxicity index as hinted by the authors, but this is not really elaborated on any further. I recommend adding this aspect to the discussion of this compound by comparing clinically available data with this compound and selected pan-HDAC inhibitors.

7. Figure 1c demonstrates that the viability of the total cell population after day 6 of viral infection is rapidly decreasing. Was the viability of the mC cell population after 7 days (moment of reactivation) also reduced and if so, how does this impact latency and the subsequent reactivation of HIV.

---

## [Author Response]

Essential revisions:1. The results show that the ratio of mC cells tot total infected cells increased over prolonged culturing in CD4 T cells. However, as the mC signal also dropped 10 fold. Additionally, there seems to be cells that solely transcribe the productive reporter.The drop in the latent reporter needs to be explained. My understanding is that this reporter should be continuously transcribed. A loss of signal of the latent reporter is a known problem with dual reporter viral systems. Why does this system show less efficiency in primary cells compared to cell lines.

We thank the reviewers for the helpful comment. In Figure 1B, as described by the reviewers that the ratio of mC cells to total infected cells increased over prolonged culturing in CD4^+^ T cells, but DP cells to total infected cells decreased. We think this phenotype can be attributed to two factors: 1) the productive infected DP cells are more prone to apoptosis caused by the cytotoxicity of HIV-1 proteins; 2) a portion of survived DP cells gradually transform to mC cells because HIV-1 transcription is turned off during the culturing process, this is also evidenced by previous works on primary cell models of HIV-1 latency, as some infected cells gradually lost the fluorescent signal of the reporter viruses (Yang et al., JCI 2009; 119: 3473–3485, He et al., *eLife* 2019; 8: e46181 and Scandella et al., Scientific Reports 2020; 10: 14642). To further verify these DP cells are more prone to apoptosis, we monitored the percentage of apoptotic cells after DFV-B infection by Annexin V and Zombie (Cat#423105, Biolegend) staining. We found that DP cells had significantly higher proportion of Annexin V-positive cells than mC cells on 3-7 days post-infection (Author response image 1), which clearly demonstrated that DP cells contained more early apoptotic cells, that contributed to the decreased ratio of DP cells in the total infected cell population.

**Author response image 1. sa2fig1:** The expression of apoptotic markers on DP and mC cells during post-infection culturing of primary CD4^+^ T cells. (A) The expression of apoptotic markers on DP and mC cells at 3 days post-infection. Flow cytometry chart shows the representative measurement of annexin V and zombie in DFV-B infected primary CD4^+^ T cells. (B) Data from Day 3, 5, 7 post-infection are displayed graphically on each indicated time point. Histograms show the percentage of positive cells. Error bars represent SDs of three technical replicates. p-values calculated using unpaired t test. ns, not significant; *p < 0.05, **p < 0.01.

In Figure 3E, we showed that the level of mC cells dropped 10-fold from 3.09% to 0.33% as concerned by the reviewers. We apologize for not stating it clear enough in the original manuscript. In fact, this data here was not generated by one-time infection and follow-up (such as Figure 1B), but was generated by multiple independent infections administered at different indicated time points, which was not the case of loss of signal of the latent reporter. The purpose here was to further verify whether HIV-1 latency was preferentially established in EMT CD4^+^ T cells, which our previous work had reported (Shan et al., Immunity 2017; 47: 766-775). Our data here showed that when the infection was administered at EMT stage (Day 6, 8, 10 after αCD3/28 stimulation), although the overall infectivity decreased, the percentages of directly generated latent infection (mC cells) significantly increased. This was demonstrated by the ratio of mC cells to total infected cells in Figure 3F, which was increased from 15.7% (Day 0 infection) to 64.8% (Day 10 infection). Our experiment confirmed that HIV-1 latency was preferentially established in EMT CD4^+^ T cells.

Additionally, regarding the potential problem of loss of signal of the latent reporter, we indeed observed that a tiny portion of single GFP^+^ cells existed after DFV-B infection (Figure 1B and Figure 3E). We traced and found that the proportion of single GFP^+^ cells decreased from 0.7% to 0.2% (similar to uninfected cells) as the culture time increased (updated Figure 2—figure supplement 1C), which is significantly lower than the previous reported dual reporter viral systems (Calvanese et al., Virology 2013; 446: 283–292 and Kim Y et al., Journal of the International AIDS Society 2019; 22: e25425). Meanwhile, the integrated DNA levels of GFP and mCherry in this single GFP^+^ cells were not different (updated Figure 2—figure supplement 1D), suggesting both reporters were still stably integrated. We speculated that this unlikely phenotype of single GFP^+^ cells could be due to the differential dynamics of fluorescent protein degradation. We agree with the reviewer’s comment about the problem of potential loss of signal of the dual reporter viral systems, we have since added additional words (lines 287-294 in revised manuscript) in the Discussion section to address.

2. Figure 2: I suggest including the PMA/ionomycin stimulation of DN and DP cells in the actual Figure 2 (currently in the supplement) as these are good controls and add further credibility to the conclusion.

We thank the reviewers for the insightful suggestion. We have modified Figure 2 by adding the PMA/ionomycin stimulation of DN and DP cells (Figure 2B). This data showed DP and DN cells were not affected by PMA/ionomycin stimulation, but more than half of the mC cells were reactivated, again suggesting that infection of DFV-B in primary CD4^+^ T cells could directly identify live latently infected cells.

3. Figure 4: The process of model generation is not quite clear from text and Figure 4. Please further clarify identification of latently infected (mCherry) and DP cells. In that process I recommend incorporating the gating plots from the supplementary figure to further clarify gating strategy and how this is different to the sorting plots shown in Figure 4B. It is also not clear at which time points in the model (shown in Figure 4B) cells were sorted and then subjected to the various stimuli.

We apologize for not stating it clear enough in the original manuscript about the process of J-mC cells generation. We have adopted the reviewer`s suggestion to show gating strategy in the modified Figure 4B. To clarify the process of J-mC generation, firstly, Jurkat cells were infected by DFV-B incorporating a VSVG envelope at 250 ng virus per million cells on Day 0 in CFM. At Day 7, the single mCherry positive cells were sorted according to the red-gated plots in Figure 4B, in which the purity of the single mCherry positive cells was greater than 99%. Next, the cells were cultured for another 30 days in CFM. At Day 37, 23.9% of the mC cells were converted to productive cells, probably due to spontaneous activation. At the same time, a small dose of TNF-α (5 ng/ml) was added to the culture and maintained for 7 days to further activate those unstable cells that were probably in the borderline state of latency. More than 70% of mC cells were reactivated at day 44. Meanwhile, the second sorting of single mCherry positive cells were performed, and subsequently the third and fourth sorting of single mCherry positive cells were performed at Day 58 and 72. J-mC cells were finalized at Day 86. Each sorting strategy was performed by the red-gated plots in Figure 4B. Besides, we have updated the experimental procedure in “Generation of J-mC cells” in the Materials and methods (lines 432-443 in revised manuscript).

4. The authors mention that previous J-Lat models are disadvantaged due to their identical integration sites, but that their J-mC model of DFB-V infection should harbour diverse integration sites. Yet, no data are presented to confirm this assumption.

We totally agree with the reviewers that this is an important point, so we have added a series of new data (the updated Figure 5) to show J-mC cells indeed harbored diverse integration sites. Briefly, we performed linear amplification–mediated high-throughput genome-wide sequencing (LAM-HTGTS) (Hu et al., Nature Protocol 2016; 11: 853-871) for HIV-1 integration site analysis on J-mC cells. We first confirmed that LAM-HTGTS was feasible for HIV-1 integration sites analysis by applying the method on J-Lat 8.4 and J-Lat 15.4 cells (updated Figure 5A and 5B). For J-mC cells, 30 days after they were obtained via multiple rounds of sorting, total DNA of the obtained J-mC cells was extracted with tissue DNA kit (OMEGA) and the genomic DNA was sheared by sonication, and subjected to LAM-HTGTS. Our data clearly demonstrated that the integration landscape of latent HIV-1 in J-mC cells was highly diverse, which represents a major strength of this cell model of HIV-1 latency comparing to the other existing cell line models. Updated descriptions of the new data have been added in the main text (lines 199-224 in revised manuscript).

5. The use of the developed model systems in the identification of ACY-1215 isinteresting and relevant, but several important pieces of information are missing, which will further strengthen the presentation of these data:a. The dose-dependent activation of GFP expression with increasing concentration of ricolinistat looks consistent and credible, but information on how long cells were stimulated for should be included in the figure or figure text, as should the utilised concentration of SAHA. It would also strengthen the interpretation if the authors could relate the used concentrations of ricolinistat to therapeutic levels during clinical dosing, ie are these levels physiologically relevant?

We thank the reviewers for pointing out this important issue. We have updated Figure 6 and added concentrations of SAHA and ACY-1215 used in the experiments. Briefly, 1 μM SAHA and 3 μM ACY-1215 were used in Figure 6 and the other supplementary figures. The recommended clinical dose of SAHA and ACY-1215 were 200-600mg once daily (QD) for 28 days (Richardson et al., Leuk Lymphoma 2008;49(3): 502–507 and Siegel et al., Blood Cancer J. 2014;4: e182) or 160 mg twice daily (BID) for 28 days (Vogl et al., Clin Cancer Res 2017; 23(13): 3307–3315 and Amengual et al., The Oncologist 2021; 10.1002/onco.13673), respectively. For in vitro system, ACY-1215 was administered at concentrations of 1–10 μM (Carew et al., Blood Advances 2018; 3:1318-1329 and Chneg et al., Biomedicine and Pharmacotherapy 2018; *109*, 2464–2471). Our application of 3 μM ACY-1215 here was in line with the other studies and was in the achievable range for clinical trials.

b. Figure 5G: Given the significant variation among individuals in cell-associated HIV RNA in the absence of stimulation, it would useful to also display the actual measured levels across DMSO, PMA/ionomycin and ACY conditions in Figure 5G. Also, which concentration of ACY was used here and is this therapeutically relevant? Finally, how many isolated CD4^+^ T cells were analysed per condition?

We thank the reviewers for this helpful suggestion. We have now added the actual measured levels of cell-associated HIV-1 RNA in the updated Figure 6G, in which the absolute copies were quantified across DMSO, PMA/ionomycin and ACY-1215 treatments. The original Figure 5G has been changed to the updated Figure 6H. 3 μM of ACY-1215 was used in the experiment and 1 million of purified CD4^+^ T cells from each ART-treated patient were used per condition. The therapeutically relevant concentration for ACY-1215 was addressed in the response above.

6. Depending on the above clarifications with regard to the most relevant concentration of ricolinistat for ex vivo and in vitro use, the compound looks like it could have similar potency to vorinostat, which is not overly impressive and may not offer any advantage compared to previously tested LRAs. One difference could be its HDAC6 selectivity, which may offer a better effect/toxicity index as hinted by the authors, but this is not really elaborated on any further. I recommend adding this aspect to the discussion of this compound by comparing clinically available data with this compound and selected pan-HDAC inhibitors.

We thank the reviewers for the insightful suggestion. We have adopted this suggestion and added another paragraph in the Discussion section to elaborate this important aspect (lines 341-358 in revised manuscript).

7. Figure 1c demonstrates that the viability of the total cell population after day 6 of viral infection is rapidly decreasing. Was the viability of the mC cell population after 7 days (moment of reactivation) also reduced and if so, how does this impact latency and the subsequent reactivation of HIV.

We thank the reviewers for the comment. The viability shown in Figure 1C was monitored in primary CD4^+^ T cells, which were previously stimulated by αCD3/CD28. In the absence of IL-2 supplement, it was expected to see the viability of the infected cells decreased as the culturing time increased. As we stated in the response to Question 1 above, primary mC cells were significantly less apoptotic than DP cells, which also suggested that the cessation of HIV-1 protein production in mC cells improved survival. To further address the reviewer’s concern, we tested the viability of the mC, DP and DN cells at 12h post-sorting, 3 days post-DMSO and 3 days post-PMA treatment. As shown in the updated Figure 2—figure supplement 1B, the viabilities of DP, DN and mC cells were comparable between DMSO and PMA treatment, and the cells got activated by the stimulation of PMA, an indication for functional cellular response. The new data in the updated Figure 2B also demonstrated that the latency reversal effect was not affected by the viabilities of DP, DN and mC cells. Taken together, the viability of the mC cells should have minimal effect on latency and the subsequent reactivation of HIV-1.